# Fermented dairy foods consumption and depressive symptoms: A meta-analysis of cohort studies

**Yupeng Luo[1]◉, Zhuo Li[2]◉, Liya Gu[2], Kui Zhang** **[2]***

**1** West China Hospital, Sichuan University, Chengdu, Sichuan, People's Republic of China, **2** Department of Forensic Pathology, West China School of Basic Medical Sciences & Forensic Medicine, Sichuan University, Chengdu, Sichuan, People's Republic of China

◉ These authors contributed equally to this work.
* cheungque@163.com

## Abstract

### Background

The gut-brain axis has been potentially proposed as a link between the intake of fermented dairy foods and depression. We carried out this meta-analysis on published cohort studies to estimate the overall depression risk of fermented dairy foods intake.

### Methods

We searched the CNKI (China National Knowledge Infrastructure) and PubMed databases for all articles within a range of published years from 2010 to 2022 on the association between fermented dairy foods intake and depression.

### Results

Finally, 8 studies met the inclusion criteria for this study, with 83,533 participants. Overall, there was statistical evidence of significantly decreased depression risk was found to be associated with fermented dairy foods intake (OR = 0.89, 95% CI = 0.81–0.98). In subgroup analysis, cheese and yogurt consumptions were significantly associated with decreased depression risk (OR = 0.91, 95% CI = 0.84–0.98 for cheese and OR = 0. 84, 95% CI = 0.72–0.99 for yogurt). However, we failed to find superabundant intake of fermented dairy foods intake decreased the risk of depression.

### Conclusion

Our meta-analysis indicated that fermented dairy foods intake may have potential beneficial effect on depression via the gut-brain axis.

**Data Availability Statement:** All relevant data are within the paper.

**Funding:** The authors received no specific funding for this work.

**Competing interests:** The authors have declared that no competing interests exist.

## Introduction

The gut-brain axis, known as a bidirectional neurohumoral communication system, which integrates the host gut and brain activities [1], has aroused public concerns [2–6]. Accumulating evidence suggests that symbiotic microorganisms, especially the microbiota which reside within the gastrointestinal system, may influence programming of social behaviors and neurodevelopment among different animal species [7]. Certain beneficial live bacteria can lead to notable improvements to neurodevelopmental disorders both in early life and in adulthood [8–10], and which suggested that supplementation of the gut microbiota may be a tractable strategy for developing novel therapeutics for complex central nerves system (CNS) disorders [11]. Several nutritional interventions have been demonstrated to influence the gut-brain axis, with probiotics confer health benefits and received particular attentions. Therefore, this axis has been proposed as an underlying link between diet quality and depression [12].

Depression is a common global mental disorder, which harms human physical and mental health seriously [13]. The estimated lifetime risk of a major depressive disorder (MDD) approaches 30% in the United States [14]. A diagnosis of depression is associated with more than 50% the incidence of suicide [15], which has been increasing and is the 10th leading cause of death in the United States [16]. The world health organization (WHO) ranked the MDD as the third leading cause of disability worldwide in 2015 [17], and projected that MDD will be ranked as the first leading cause of disability by the year of 2030 [18]. However, the pathophysiology of depression remains incompletely understood [15].

Fermented foods are generally made through controlled microbial growth and enzymatic conversions of major and minor food components [19]. Yogurt and cheeses are the main products produced by specific microorganisms or complex microbial communities for fermentation from different types of milk [20]. That relies on the use of the addition probiotic candidates [21, 22]. Previous studies have shown beneficial contribution of fermented dairy foods to cardiovascular diseases [23, 24], type 2 diabetes [23], and cancer [25], but failed to support the use of fermented dairy foods interventions for cognitive outcomes [26].

The associations of risk relating to depression with fermented dairy foods intake have already been widely studied [27–34]. However, the results remained inconsistent. Thus, we carried out this meta-analysis on all published cohort studies to detect the overall effects from a quantitative synthesis of the accumulated data to estimate the overall depression risk of fermented dairy foods intake. This study was reported in accordance with the PRISMA statement for reporting systematic reviews and meta-analysis [35].

## Methods

### Publication search and inclusion criteria

We searched the CNKI (China National Knowledge Infrastructure) and PubMed databases for all articles within a range of published years from 2010 to 2022 on the association between fermented dairy foods intake and depressive symptoms (last search was update September 29, 2022). The following terms were used in this search: 'fermented food or fermented milk or cheese or yogurt or cultured milk or lactic acid bacteria' and 'depressive or depression'. The references cited in the research papers were also scanned to identify the relevant publications. Combining searches resulted in 719 abstracts (Fig 1). An additional 2 studies were identified through review articles, for a total of 721studies.

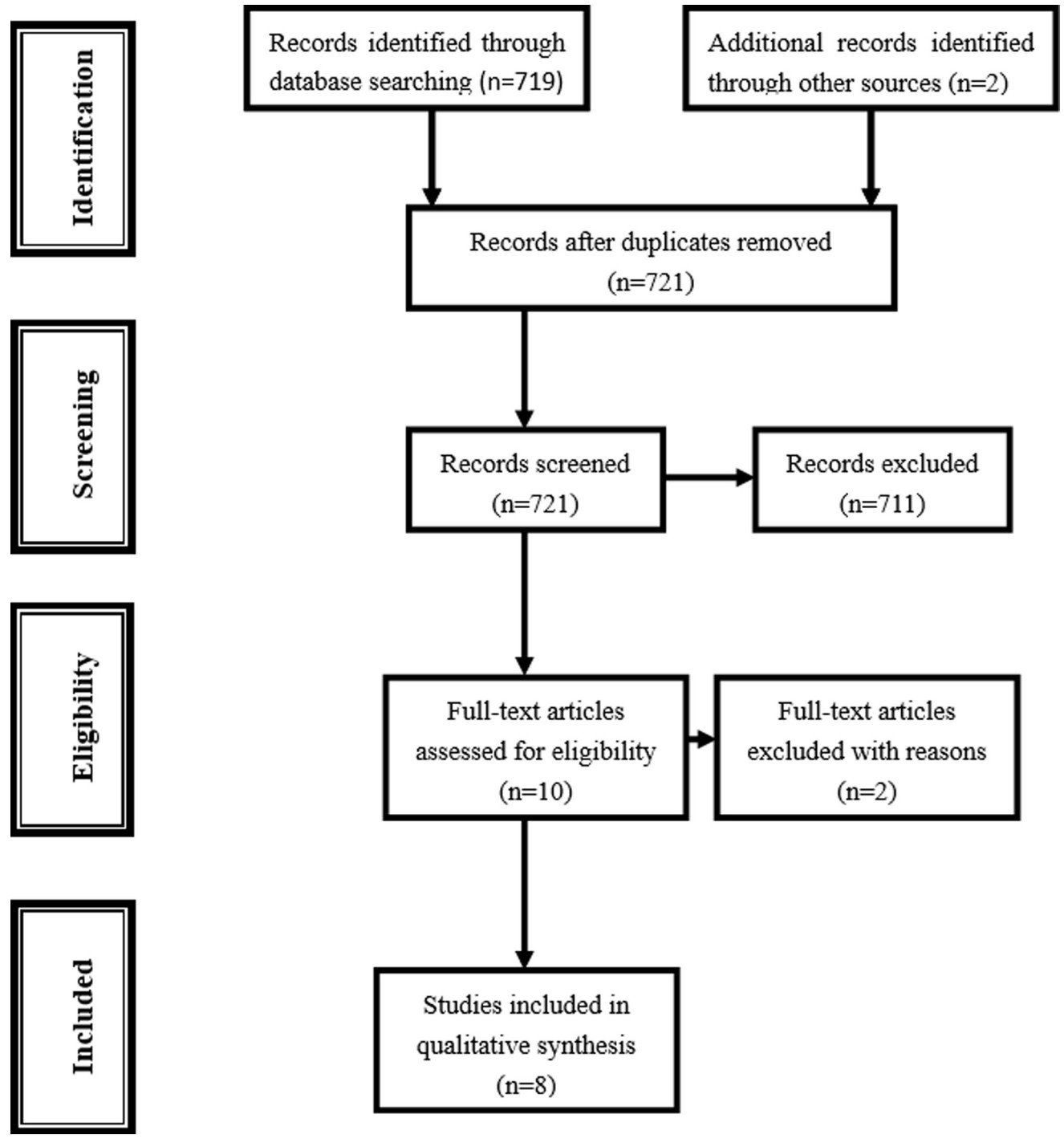

**Fig 1. Flowchart for identification of studies.**

We evaluated the eligible studies if all the following conditions were met: (1) evaluation of the association of depression risk with yogurt or cheese consumption; (2) diet was generally assessed by food frequency questionnaire; (3) ORs and 95% CIs can be acquired or calculated from sufficient data in the manuscript or supplementary materials; (4) the publication was a cohort study; and (5) the study was published in English.

### Data extraction

Two authors (Yupeng Luo and Zhuo Li) independently reviewed and extracted the data needed. Disagreements were resolved through discussion among the authors to achieve a consensus. Publications were read by Yupeng Luo in order to check original data extraction. The following information were recorded from each study: first author, year of publication, follow-up period, region, sex, dairy type, dietary assessment, number of cases, number of controls (all of the data are shown in Table 1).

### Statistical analysis

The association between fermented dairy foods intake and depression risk was assessed with the odds ratio (OR) corresponding to the 95% confidence interval (95% CI). Stratified analyses was performed to assess the association between cheese, and yogurt intake and depression risk.

The statistical heterogeneity among studies was assessed with the Q-test and $I^2$ statistics [36]. If no obvious heterogeneity, the fixed-effects model (the Mantel-Haenszel method) was used to estimate the summary OR [37]; otherwise, the random-effects model (the DerSimonian and Laird method) was used [38]. Finally, random effects models were selected to calculate the overall OR estimates and 95% CIs by comparing the highest intake versus the lowest consumption of fermented dairy foods, and comparing the higher consumption versus lower intake of fermented dairy foods. To explore sources of heterogeneity across studies, we did logistic meta-regression analyses. We examined the following study characteristics: publication year, follow-up period, region, dairy type, diagnostic criteria of depression, number of cases, morbidity of cases, number of controls, and morbidity of controls. Publication bias was evaluated with funnel plot and Begg's rank correlation method [39]. The statistical analyses were performed by STATA 12.0 software (Stata Corp., College Station, TX).

## Results

### Characteristics of studies

Out of a total of 721 abstracts were screened, 10 were retrieved for more detailed evaluation. Of the two excluded studies, one papers had nothing on depression [40], and one papers lacked sufficient data [41] (shown in Fig 1). Finally, 8 cohort studies met the inclusion criteria for this study [27–34], with 83,533 participants, 6,119 depression. The details of including first author, year of publication, region, follow-up period, dairy type, sex, diagnostic criteria of depression, dietary assessment, number of cases, number of controls, and cut off value of comparison in the selected studies were listed in Table 1.

### Quantitative synthesis

The evaluation of fermented dairy foods intake and depression risk was shown in Table 2 and Fig 2. Overall, there was statistical evidence of significantly decreased depression risk was found to be associated with fermented dairy foods intake (OR = 0.89, 95% CI = 0.81–0.98). In subgroup analysis, cheese and yogurt consumptions were significantly associated with decreased depression risk (OR = 0.91, 95% CI = 0.84–0.98 for cheese and OR = 0. 84, 95% CI = 0.72–0.99 for yogurt). However, we failed to find superabundant intake of fermented dairy foods intake decreased the risk of depression.

### Evaluation of heterogeneity

There was heterogeneity among studies in overall comparisons ($P_{heterogeneity} < 0.001$, $I^2 = $ 67.8%, $Tau^2 = 0.0222$ for higher intake, $P_{heterogeneity} = 0.001$, $I^2 = 63.0\%$, $Tau^2 = 0.0276$ for

**Table 1. Characteristics of literatures included in the meta-analysis.**

| Reference | Region | Follow up time (year) | diagnostic criteria of depression | Dairy type | Gender | No. of cases | No. of subjects | Cut off value | | Adjusted factors |
|---|---|---|---|---|---|---|---|---|---|---|
| | | | | | | | | Higher intake | Highest intake | |
| Sun J [32] | China | 10 | PHQ-9 | Yogurt | Both | 1,971 | 19,952 | Any intake | ≥183.75g/day | age, gender, race, educational level, marital status, family income, BMI, work physical activity, recreational physical activity, alcohol consumption, smoking status, diabetes, hypertension, heart disease, stroke and total daily energy intake |
| | | | | Cheese | | | | Any intake | ≥42.00g/day | |
| Takahashi F [33] | Japan | 2.8 | K6 | Yogurt | Women | 280 | 9,030 | ≥60g/day | | parity, body mass index, marital status, number of children, mood after pregnancy, history of mental health disorders, age, academic history, job type, household income, present smoking status |
| | | | | Yogurt drink | | | | Any intake | ≥80ml/day | |
| | | | | Cheese | | | | Any intake | ≥5g/d | |
| Yu B [34] | China | 9 | SDS | Yogurt | Both | 1,290 | 19,596 | Any intake | ≥twice/day | age, sex, BMI, smoking status, drinking status, physical activity, marital status, total energy intake, household incomes, occupations, educational levels, social contact, cohabitants, metabolic syndrome, and milk consumption |
| Miyake Y [28] | Japan | 1 | CES-D | Yogurt | Women | 337 | 1,745 | Lowest quartile | Highest quartile | age, gestation, region of residence, number of children, family structure, history of depression, family history of depression, smoking, secondhand smoke exposure at home and at work; job type, household income, education, body mass index, and intake of saturated fatty acids, eicosapentaenoic acid plus docosahexaenoic acid, and vitamin D |
| | | | | Cheese | | | | Lowest quartile | Highest quartile | |
| Perez-Cornago A [30] | Spain | 9.3 | DSMMD | Yogurt | Both | 727 | 14,539 | ≥63g/week | ≥875g/week | age, sex, smoking, physical activity, total energy intake, baseline BMI, living alone, unemployment, marital status, and personality traits |
| Miyake Y [29] | Japan | 1 | CES-D | Yogurt | Women | 108 | 1,319 | Lowest quartile | Highest quartile | age, gestation, region of residence, number of children, family structure, history of depression, family history of depression, job type, education, body mass index, having smoked during pregnancy, cesarean delivery, baby's sex, baby's birth weight, and total energy intake |
| | | | | Cheese | | | | Lowest quartile | Highest quartile | |
| Mahdavifar B [27] | Iran | 3 | DASS 21 | Yogurt | Both | 578 | 7,387 | Lowest tertile | Highest tertile | Age, energy intake, smoking status, physical activity, marital status, educational levels, chronic disease, residential status, occupation, BMI |
| | | | | Cheese | | | | Lowest tertile | Highest tertile | |
| | | | | Yogurt drink | | | | Lowest tertile | Highest tertile | |
| Sangsefidi ZS [31] | Iran | 3 | DASS 21 | Yogurt | Both | 827 | 9,965 | Any intake | >2 glasses/week | age, education level, physical activity level, history of chronic diseases, smoking and BMI |

PHQ-9, the Patient Health Questionnaire; K6, the Kessler 6-item psychological distress scale; SDS, the Self-Rating Depression Scale; CES-D, the Epidemiologic Studies Depression Scale; DSMMD, the Diagnostic and Statistical Manual of Mental Disorders; DASS 21, depression, anxiety, and stress scale 21.

highest intake). To explore sources of heterogeneity across studies, subgroup analysis was performed ($P_{heterogeneity} < 0.001$, $I^2 = 76.7\%$, $Tau^2 = 0.0365$ for higher yogurt intake, $P_{heterogeneity} = 0.532$, $I^2 = 0\%$, $Tau^2 < 0.001$ for higher cheese intake; $P_{heterogeneity} < 0.001$, $I^2 = 76.9\%$, $Tau^2 = 0.0615$ for highest yogurt intake, $P_{heterogeneity} = 0.110$, $I^2 = 46.9\%$, $Tau^2 = 0.0153$ for highest

**Table 2. Associations between fermented dairy foods and risk of depression.**

| | All | | | | Cheese | | | | Yogurt | | |
|---|---|---|---|---|---|---|---|---|---|---|---|
| | N[a] | Case/control | OR (95%CI) | P[b] | N[a] | Case/control | OR (95%CI) | P[b] | N[a] | Case/control | OR (95%CI) | P[b] |
| Higher intake | 15 | 6,119/83,533 | **0.89(0.81–0.98)** | 0.0222 | 5 | 3,275/39,433 | **0.91(0.84–0.98)** | <0.001 | 8 | 6,119/83,533 | **0.84(0.72–0.99)** | 0.0365 |
| Highest intake | 14 | 6,119/83,533 | 0.92(0.82–1.03) | 0.0303 | 5 | 3,275/39,433 | 0.86(0.73–1.01) | 0.0153 | 7 | 5,839/74,503 | 0.96(0.78–1.20) | 0.0615 |

[a] Number of comparisons.

[b] P value of Q-test for heterogeneity test.

Boldfaced values indicate a significant difference at the 5% level.

cheese intake). Furthermore, we assessed publication year, region, follow-up period, disease type, and dairy type, number of cases, number of controls, morbidity of cases, and morbidity of controls. Finally, logistic meta-regression analyses revealed that morbidity of cases and morbidity of controls could substantially influence the initial heterogeneity.

## Sensitivity analysis

Sensitivity analysis was performed to find the influence of a single study on the overall meta-analysis estimate by omitting one study at a time, and the omission of any study made no significant difference, indicating that our results were statistically reliable.

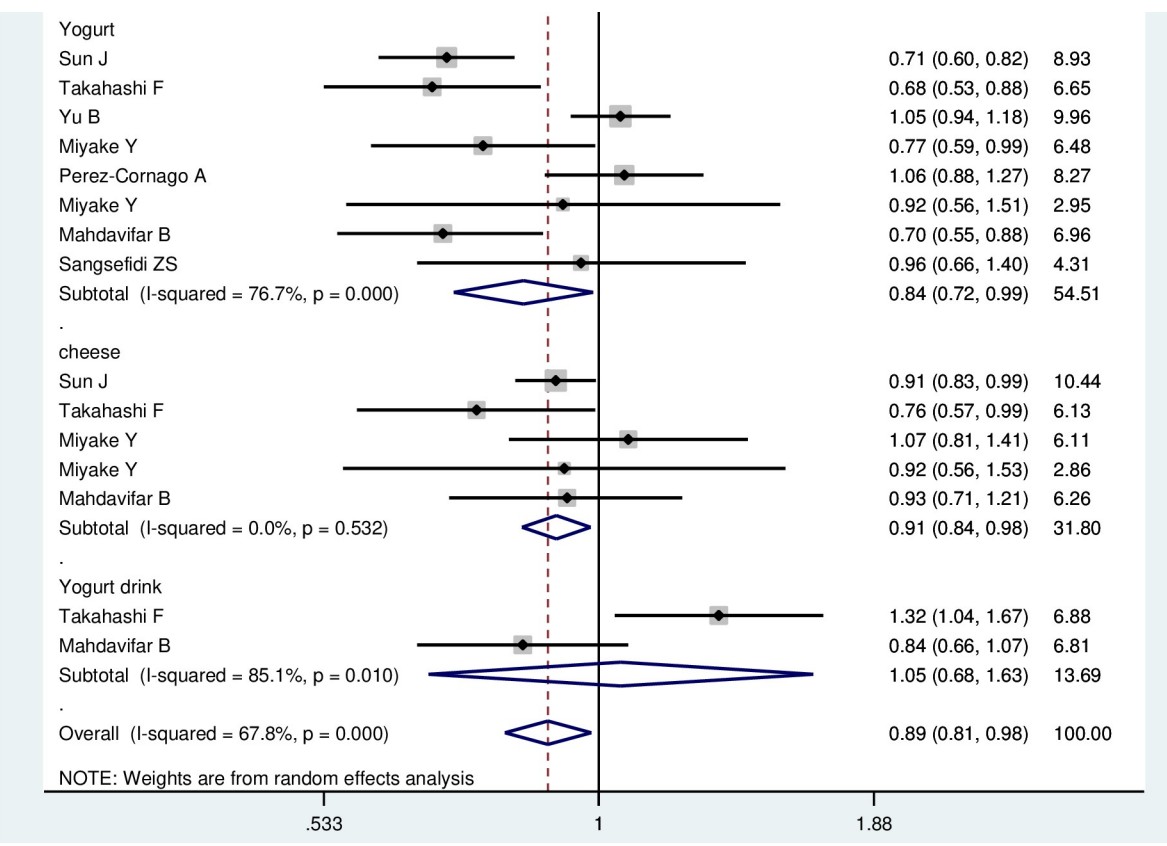

**Fig 2. Forest plot of depression risk associated with fermented dairy foods.**

### Publication bias

The Begg's test was performed to evaluate the publication bias of selected cohort literatures. Fig 3 displays a funnel plot that examined the fermented dairy foods and depression risk included in the meta-analysis. No evidence of publication bias in our study was observed ($P$ = 0.843).

### Discussion

In the present study, we carried out this meta-analysis on 8 published cohort studies with 83,533 participants to estimate the overall depression risk of fermented dairy foods. Overall, statistical evidence of significantly decreased depression risk was found to be significantly associated with fermented dairy foods intake (OR = 0.89, 95% CI = 0.81–0.98). In subgroup analysis, cheese and yogurt consumptions were significantly associated with decreased depression risk (OR = 0.91, 95% CI = 0.84–0.98 for cheese and OR = 0. 84, 95% CI = 0.72–0.99 for yogurt). However, we failed to find superabundant intake of fermented dairy foods intake decreased the risk of depression.

Fermented dairy foods, with abundant probiotics, have possible beneficial effects on mental health, and growing evidence supporting the role of probiotics in regulating the brain function and subsequent emotional behavior. Recently, new definitions have been added to the probiotic terminology as parabiotics and postbiotics [42]. Evidence suggests that probiotic can modulate brain function by modifying the gut microbiota, decreasing low-grade inflammation, and influencing the production of neurochemicals [12, 43]. Preclinical evaluation suggests that manipulation of the gut microbiota with specific probiotics can influence the depression-like behaviors [44, 45]. Potential mechanisms for these effects may include that probiotics can reduce gut inflammation, decrease intestinal permeability, reduce stress responses via the

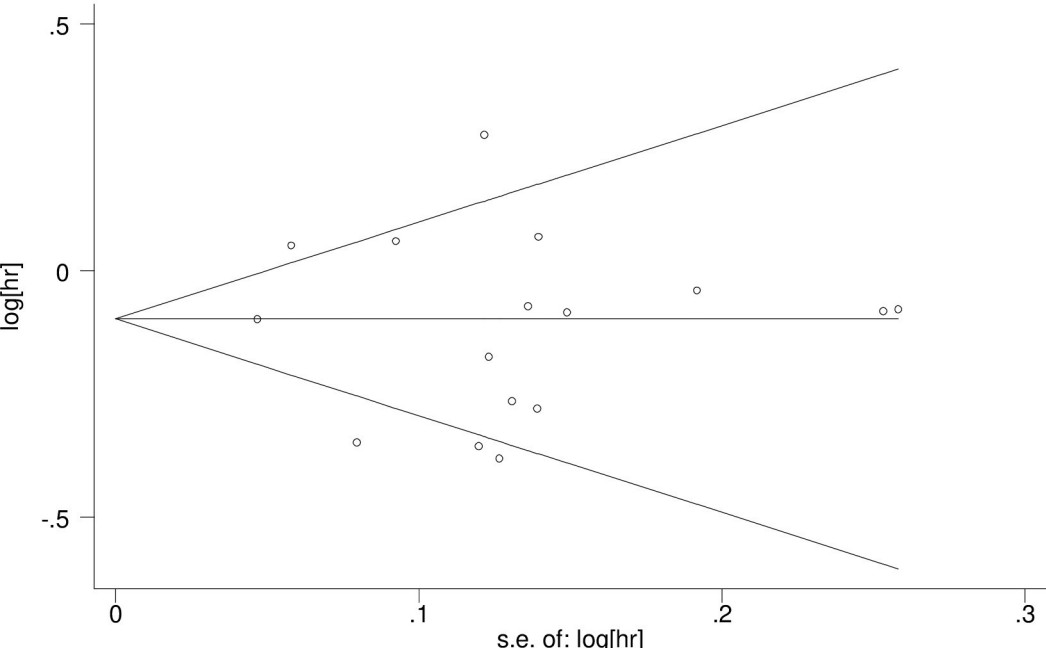

**Fig 3. Begg's funnel plot for publication bias test.** Each point represents a separate study for the indicated association. s.e: standardized effect.

hypothalamic–pituitary–adrenal axis, and alter GABA receptor expression [46, 47]. A brain imaging study has shown that brain activity in a network of brain areas involving processing negative emotional facial expressions can be reduced in person who consumed the fermented milk product containing a combination of probiotics [48]. Another recent study revealed that significant decreased depression score was found in female intake of probiotic supplementation [49]. Therefore, the gut-brain axis has been proposed as an underlying link between fermented dairy foods and depression.

Interestingly, we failed to find superabundant intake of fermented dairy foods intake decreased the risk of depression. Previous research found that superabundant intake of fermented dairy foods even increased the risk of depressive symptoms dramatically [34]. However, there is no persuasive evidence for any negative impact of probiotics on mental health, a possible explanation is that something other than the probiotics might account for this adverse result, and the pivotal concern might be the added sugar or sweetener. Epidemiologic studies have suggested a positive association between consumption of sweets and depression [50, 51]. And the mechanism have been assumed as that high sucrose diet has been suggested to reduce brain glucose uptake, induce psychosis-related behavioral endophenotypes and microcapillary impairments, which may cause psychiatric disorders [52].

A few limitations of our study should be considered. Although we did not observe significant publication bias, publication bias may be possible in any meta-analysis. Moreover, the wide variation in bacterial species and strains, different consumed products, and the potential inhomogeneity of dietary culture may have possible effect on the results. Finally, lack of unified cutoff value for the fermented dairy foods consumption in the enrolled studies may also potentially affect the results.

In conclusion, our meta-analysis indicated that fermented dairy foods intake may have potential beneficial effect on depression via the gut-brain axis. Moreover, further studies estimating the underlying functional effect may eventually provide a better, comprehensive understanding.

## Supporting information

**S1 Checklist. PRISMA 2020 checklist.**
(DOCX)

## Author Contributions

**Conceptualization:** Kui Zhang.

**Data curation:** Yupeng Luo, Zhuo Li, Liya Gu.

**Methodology:** Yupeng Luo, Zhuo Li.

**Writing – original draft:** Yupeng Luo, Zhuo Li, Kui Zhang.

**Writing – review & editing:** Liya Gu.

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
