## [Decision Letter · Decision Letter 0]

12 Dec 2022

PONE-D-22-29070Fermented dairy foods consumption and depressive symptoms: a meta-analysis of cohort studiesPLOS ONE

Dear Dr. Zhang,

Thank you for submitting your manuscript to PLOS ONE. After careful consideration, we feel that it has merit but does not fully meet PLOS ONE’s publication criteria as it currently stands. Therefore, we invite you to submit a revised version of the manuscript that addresses the points raised during the review process.Please pay particular attention to the suggestins by reviewer 2, regarding the information on the meta analysis. 

We look forward to receiving your revised manuscript.

Kind regards,

Guadalupe Virginia Nevárez-Moorillón, Ph.D.

Academic Editor

PLOS ONE

Journal Requirements:

Reviewers' comments:

Reviewer's Responses to Questions

**Comments to the Author**

1. Is the manuscript technically sound, and do the data support the conclusions?

Reviewer #1: Yes

Reviewer #2: Yes

2. Has the statistical analysis been performed appropriately and rigorously? 

Reviewer #1: Yes

Reviewer #2: Yes

3. Have the authors made all data underlying the findings in their manuscript fully available?

Reviewer #1: Yes

Reviewer #2: No

4. Is the manuscript presented in an intelligible fashion and written in standard English?

Reviewer #1: Yes

Reviewer #2: Yes

5. Review Comments to the Author

Reviewer #1: Dear Authors, here are my comments on your work, "Fermented dairy foods consumption and depressive symptoms: a meta-analysis of cohort studies." This work performed a meta-analysis of all published cohort studies to estimate the overall depression risk of fermented dairy food intake. The introduction, methodology, and results were well-written and easy to follow.

Specific comments

Line 20, please define the WHO acronym.

Please describe what you mean by "highest fermented dairy products." (describe the type of dairy products considered as highest fermented dairy products)

In the section "Evaluation heterogeneity," please clarify or discuss the meaning of I2 and Tau2.

In the section "Sensitivity analysis," please include the p-value for the lack of significance.

Why did you include data from fermented soybeans products (Table 1)? When this product is not a fermented dairy product. Please exclude this information from your meta-analysis.

Reviewer #2: The current study used a meta-analysis to look into the link between fermented dairy and the risk of depression.

The authors declared that they included studies that assessed yogurt and cheese intake, but they also included a study with fermented soybeans. 

The exclusion criteria and PICO framework did not note.

The Eggers test can be used to check for publication bias.

The quality of the included studies should be assessed.

The authors should explain the cut-off values for higher intake and highest intake in more detail.

Please add the results of the meta-regression.

Why were yogurt and yogurt drinks reported separately in the forest plot?

Adjusted variables and age are added to Table 1.

There are four duplicate studies in the present meta-analysis. The recent study with a larger sample size should be selected and included in the meta-analysis.

6. PLOS authors have the option to publish the peer review history of their article (what does this mean?). If published, this will include your full peer review and any attached files.

Reviewer #1: **Yes: **Nestor Gutiérrez-Méndez

Reviewer #2: No

---

## [Author Response · Author response to Decision Letter 0]

18 Jan 2023

Dear Editor,

Thank you for your letter dated December 13, 2022 concerning our manuscript entitled “Fermented dairy foods consumption and depressive symptoms: a meta-analysis of cohort studies” (Manuscript ID PONE-D-22-29070). We would like to thank the anonymous reviewers and the editor for their constructive comments on revising this manuscript. According to the comments of the reviewers, we have revised the related part in the original manuscript and the revised portions were marked in red bold. The following is the correspondence to the reviewers concerning the comments and suggestions about the manuscript.

Answers to the reviewer 1 

1. Line 20, please define the WHO acronym. 

Answer: thanks for the reviewer’s comments, we have defined the WHO acronym.

2. Please describe what you mean by "highest fermented dairy products." (describe the type of dairy products considered as highest fermented dairy products)

Answer: thanks for the reviewer’s comments, we have revised the relating sentences.

3. In the section "Evaluation heterogeneity," please clarify or discuss the meaning of I2 and Tau2.

Answer: thanks for the reviewer’s comments. To explore sources of heterogeneity across studies, subgroup analysis was performed (P heterogeneity ＜ 0.001, I2 =76.7%, Tau2 = 0.0365 for higher yogurt intake, P heterogeneity = 0.532, I2 = 0%, Tau2 ＜ 0.001 for higher cheese intake; P heterogeneity ＜ 0.001, I2 =76.9%, Tau2 = 0.0615 for highest yogurt intake, P heterogeneity = 0.110, I2 = 46.9%, Tau2 = 0.0153 for highest cheese intake). Furthermore, we assessed publication year, region, follow-up period, disease type, and dairy type, number of cases, number of controls, morbidity of cases, and morbidity of controls. Finally, logistic meta-regression analyses revealed that morbidity of cases and morbidity of controls could substantially influence the initial heterogeneity.

4.In the section "Sensitivity analysis," please include the p-value for the lack of significance.

Answer: thanks for the reviewer’s comments. The results of sensitivity analysis were listed as follows:

5.Why did you include data from fermented soybeans products (Table 1)? When this product is not a fermented dairy product. Please exclude this information from your meta-analysis.

Answer: thanks for the both reviewers’ comments, we have re-analyzed the data.

Answers to the reviewer 2 

1. The authors declared that they included studies that assessed yogurt and cheese intake, but they also included a study with fermented soybeans.

Answer: thanks for the both reviewers’ comments, we have re-analyzed the data.

2. The exclusion criteria and PICO framework did not note.

Answer: We evaluated the eligible studies if all the following conditions were met: (1) evaluation of the association of depression risk with yogurt or cheese consumption; (2) diet was generally assessed by food frequency questionnaire; (3) ORs and 95% CIs can be acquired or calculated from sufficient data in the manuscript or supplementary materials; (4) the publication was a cohort study; and (5) the study was published in English.

3. The Eggers test can be used to check for publication bias..

Answer: thanks very much for the reviewer’s suggestion, the Eggers test and Begg’s test were performed, all the value were above 0.05.

4. The quality of the included studies should be assessed.

Answer: We evaluated the eligible studies if all the following conditions were met: (1) evaluation of the association of depression risk with yogurt or cheese consumption; (2) diet was generally assessed by food frequency questionnaire; (3) ORs and 95% CIs can be acquired or calculated from sufficient data in the manuscript or supplementary materials; (4) the publication was a cohort study; and (5) the study was published in English.

5. The authors should explain the cut-off values for higher intake and highest intake in more detail.

Answer: the details of cut-off values were listed in table 1.The limitation of unified cutoff value for the fermented dairy foods consumption in the enrolled studies may also potentially affect the results was listed in the discussion.

6. Please add the results of the meta-regression.

Answer: thanks very much for the reviewer’s comment, we have added the results of subgroup analysis.

7. Why were yogurt and yogurt drinks reported separately in the forest plot?

Adjusted variables and age are added to Table 1.

There are four duplicate studies in the present meta-analysis. The recent study with a larger sample size should be selected and included in the meta-analysis.

Answer: thanks very much for the reviewer’s comment, yogurt and yogurt drinks are different type of fermented dairy foods, thus we reported it separately.

Thanks very much for the reviewer’s comment, adjusted variables are added to Table 1.

Thanks very much for the reviewer’s comment, we have searched the relating website with the inclusion criterion, and eight studies were available recently.

At this time, we address the statement that:

The authors received no specific funding for this work.

Thanks the anonymous reviewers and the editor again.

Sincerely Yours authors

December 19, 2022

---

## [Editor Report · Decision Letter 1]

23 Jan 2023

Fermented dairy foods consumption and depressive symptoms: a meta-analysis of cohort studies

PONE-D-22-29070R1

Dear Dr. Zhang,

We’re pleased to inform you that your manuscript has been judged scientifically suitable for publication and will be formally accepted for publication once it meets all outstanding technical requirements.

Kind regards,

Guadalupe Virginia Nevárez-Moorillón, Ph.D.

Academic Editor

PLOS ONE

---

## [Editor Report · Acceptance letter]

27 Jan 2023

PONE-D-22-29070R1 

Fermented dairy foods consumption and depressive symptoms: a meta-analysis of cohort studies 

Dear Dr. Zhang:

I'm pleased to inform you that your manuscript has been deemed suitable for publication in PLOS ONE. Congratulations! Your manuscript is now with our production department. 

Kind regards, 

on behalf of

Dr. Guadalupe Virginia Nevárez-Moorillón 

Academic Editor

PLOS ONE